# Phase Formation and Stabilization Behavior of Ca-PSZ by Post-Heat Treatment

**Hyunjo Yoo [1], Hwanseok Lee [2], Kanghee Jo [2], Juyoung Kim [1], Ilguk Jo [3],\* and Heesoo Lee [2],\***

[1]   School of Convergence Science, Pusan National University, Busan 609-735, Korea
[2]   School of Materials Science and Engineering, Pusan National University, Busan 609-735, Korea
[3]   School of Materials Science and Engineering, Dong-Eui University, Busan 47340, Korea
\*   Correspondence: ijo@deu.ac.kr (I.J.); heesoo@pusan.ac.kr (H.L.)

**Abstract:** The phase formation and stabilization behaviors of calcia partially stabilized zirconia (Ca-PSZ) were investigated with regard to the CaO content and post-heat treatment. Sintered specimens were prepared by adding 2, 3, 4, and 5 mol% to CaO to $ZrO_2$, and post-heat treatment were conducted. In the X-ray diffraction pattern, the monoclinic peak decreased, the tetragonal peak increased upon CaO doping, and no $CaZrO_3$ peak was observed. Transmission electron microscopy images of the Ca-PSZ showed that the d-spacing of 4CSZ $(200)_m$ extended from 0.260 nm to 0.266 nm subsequent to post-heat treatment. The coefficient of thermal expansion gradually increased in accordance with the dopant concentration, in addition, it increased even after the post-heat treatment. These results are related to the increase in tetragonal phase, which has a relatively higher coefficient of thermal expansion than that of the monoclinc phase. According to the Vickers hardness measurement, the hardness of all specimens increased gradually as the concentration of CaO increased, and the hardness of the 5CSZ was improved from 676 to 774 Hv by the post-heat treatment.

**Keywords:** calcia stabilized zirconia; thermal treatment; phase formation; thermal expansion; mechanical properties

## 1. Introduction

Research on materials that can achieve high efficiency and can be used in extreme environments are being widely investigated with the advent of the 4th Industrial Revolution. Zirconia ($ZrO_2$) has excellent mechanical properties, low thermal conductivity, and corrosion resistance to withstand harsh environments, and it is widely investigated for structural materials (mechanical materials and refractories) to functional materials (solid electrolytes for fuel cells and oxygen sensors) [1–3].

When zirconia is exposed to high temperatures for a long time, phase transformation from the tetragonal phase to the monoclinic phase occurs. This causes cracks by volume expansion, leading to a rapid reduction in strength due to intergranular fracture; nevertheless, various materials are being developed to solve this problem [4,5]. Yttria- or magnesia-stabilized zirconia shows excellent mechanical properties and chemical stability. It is used at high temperatures (~1000 °C), such as in gas turbines for power generation, oxygen sensors, and fuel cells as reported by previous studies [6–8].

Calcia partially stabilized zirconia (Ca-PSZ) has excellent properties such as thermal stability and corrosion resistance in extreme environments. Owing to these properties, Ca-PSZ is used in immersion nozzle refractories for steel continuous steel casting, which entails extremely high temperatures (1500 °C) and corrosive molten flux environments [9–12]. Higher thermal stability and corrosion resistance are required to satisfy the more severe application conditions and higher requirements of continuous casting time.

It is necessary to limit the thermal and chemical degradation of the Ca-PSZ, which is caused by the destabilization of $ZrO_2$ to resolve these problems. A typical approach to improve corrosion resistance is adding other stabilizing elements [13]. However, increasing

the doping level is disadvantageous on the mechanical properties, it is necessary to control the CaO content [14]. Ca-PSZ undergoes a preheating process at 1200 °C to receive molten steel above 1500 °C and is continuously exposed to high temperatures afterward. Therefore, variable control such as changing the content of stabilizing element and heat treatment is required to improve the thermal stability of zirconia.

In this study, the 2–5 mol% Ca-PSZ was synthesized and post heat-treatment was conducted to improve the thermal and mechanical properties. The crystal structure and phase fraction of Ca-PSZ by the post-heat treatment, including the CaO content change, were analyzed. The interplanar distances were confirmed through diffraction patterns obtained by fast Fourier transform (FFT) of the cross-section of high-resolution TEM images of the Ca-PSZ. Further, the coefficient of thermal expansion (CTE) and Vickers hardness were measured to analyze the thermal and mechanical properties.

## 2. Experimental

In this study, the Ca-PSZ was synthesized through the solid-phase synthesis using $ZrO_2$ (99%, Daejung Chemicals & Metals Co., Siheung, Korea) and CaO (98%, Junsei Chemical Co., Tokyo, Japan) as starting materials. The $ZrO_2$ powders and the CaO powders were prepared as shown in Table 1. The powders were ball-milled for 24 h in ethyl alcohol using partially stabilized zirconia balls and grinding pots. After drying, the powders were calcinated at 1000 °C for 2 h.

**Table 1.** Composition of the Ca-PSZ specimens with different amounts of CaO.

| Compound | Composition | |
| :---: | :---: | :---: |
| | CaO Addition (Mol%) | $ZrO_2$ (Mol%) |
| 2CSZ | 2 | 98 |
| 3CSZ | 3 | 97 |
| 4CSZ | 4 | 96 |
| 5CSZ | 5 | 95 |

The disk specimens (Ø 25 mm × ~2.5 mm thick) and the bar-type specimens (5 mm × 4.5 mm × 35 mm) were uniaxially pressed at 3 ton/m$^2$ in a steel die. During sintering, the samples were heated up to 1600 °C with a heating rate of 5 K/min and held at this temperature for 6 h. The samples were then cooled to the ambient temperature in air. The post-heat treated specimens were obtained by heating the sintered specimens at 1200 °C for 100 h. The X-ray diffraction (XRD) patterns of the Ca-PSZ specimens were collected at room temperature using a step scan procedure (2 h = 20–80°, step interval: 0.02°, CuK$\alpha$ radiation, Rigaku Ultima-IV, Rigaku, Tokyo, Japan). The specimens for microstructure analysis were prepared using a focused ion beam (FIB, Scios DualBeam, Los Santos, SA, USA). The phases of the synthesized specimens were identified using transmission electron microscopy (FE-TEM, JEM 2100F, Jeol, Tokyo, Japan), and dispersive spectrometry (EDS) point analysis was performed to confirm the elements in the phase. Thermomechanical analysis (TMA450, Ta Instruments) was used for analysis at temperatures from 30 °C to 900 °C at 5.0 °C/min per minute for measuring the coefficient of thermal expansion. Micro Vickers hardness tester (Wilson® VH1102, Chicago, IL, USA) was used to identify the mechanical properties of Ca-PSZ. The measurement was conducted under a maximum load of 20 N for 15 s on three parallel specimens for each composition and the test were performed at three points for each specimen.

## 3. Results and Discussion

We analyzed the XRD patterns of Ca-PSZ samples, as shown in Figure 1. All specimens showed a mixed phase of a tetragonal phase and a monoclinic phase, indicating that they were partially stabilized by the CaO, and $CaZrO_3$ peak was not observed. In the XRD pattern before and after post-heat treatment, the intensity of monoclinic phase positioned at 28.6° and 31.8° decreased, and the peaks at 30.5° corresponding to the tetragonal phase

increased according to the CaO content. This is the result of stabilization of the monoclinic phase into a tetragonal phase as Ca$^{2+}$ is dissolved into the ZrO$_2$ lattice as the amount of CaO increases. To quantify the monoclinic volume fraction of Ca-PSZ before and after post heat- treatment, the peak intensity was calculated by the Potter and Heuer equation as shown in Equations (1) and (2) [15].

$$X = \frac{I(\bar{1}11)_m + I(111)_m}{I(\bar{1}11)_m + I(111)_m + I(101)_t} \tag{1}$$

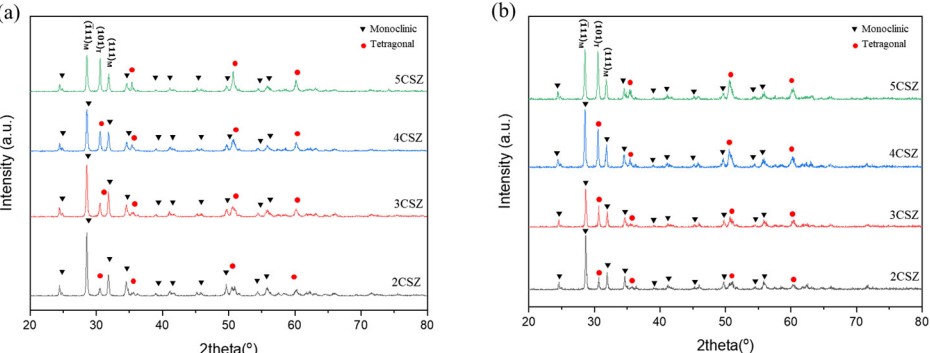

**Figure 1.** XRD pattern before (**a**) and after-heat treatment (**b**) of Ca-PSZ.

The *X* indicates the integrated intensity ratio and *I* express integrated intensity of monoclinic and tetragonal phase reflection. The volume fraction of the monoclinic phase $f_m$ is calculated by Equation (2).

$$f_m = \frac{PX}{1 + (P-1)X} \tag{2}$$

The *P* indicates intensity factor, and in the monoclinic-tetragonal ZrO$_2$ system, *p* value is 1.311. The calculated monoclinic phase volume fraction is listed in Table 2. The post-heat treatment reduced the volume fraction of monoclinic phase, and in particular, the monoclinic volume fraction of 3CSZ decreased from 84.42% before post-heat treatment to 71.70% after the treatment. These results are considered the Ca-PSZ stabilized from the monoclinic phase to the tetragonal phase by diffusion of Ca without precipitation of Ca to grain boundaries by post-heat treatment.

**Table 2.** Monoclinic phase volume fraction of Ca-PSZ.

| Compound | Monoclinic Phase Volume Fraction (%) | |
|:---:|:---:|:---:|
| | **Before-Heat Treatment** | **After-Heat Treatment** |
| 2CSZ | 91.42 | 85.47 |
| 3CSZ | 84.42 | 71.70 |
| 4CSZ | 75.25 | 67.61 |
| 5CSZ | 62.50 | 59.38 |

The TEM images and the inverse FFT images of Ca-PSZ are shown in Figure 2. The monoclinic phase was confirmed in 2CSZ, 3CSZ, and 4CSZ before and after the post-heat treatment, and the tetragonal phase was observed in 5CSZ. An increase in the lattice parameter due to CaO doping was observed through the (002)$_m$ d-spacing of the post-heat treated 2CSZ and 3CSZ (0.272 nm and 0.274 nm, respectively). The increase in the lattice parameter occurs as the Ca$^{2+}$ (1.00 Å), which has a larger ionic radius than that of Zr$^{4+}$ (0.86 Å), is substituted into ZrO$_2$ lattice. Additionally, the d-spacing of the 4CSZ (200)$_m$ plane increased from 0.260 nm to 0.266 nm after heat treatment. This indicates diffusion of the Ca$^{2+}$ from rich regions to a lean region (monoclinic phase) by the heat treatment, and the TEM-EDS results (Figure 3) before and after post-heat treatment support this result.

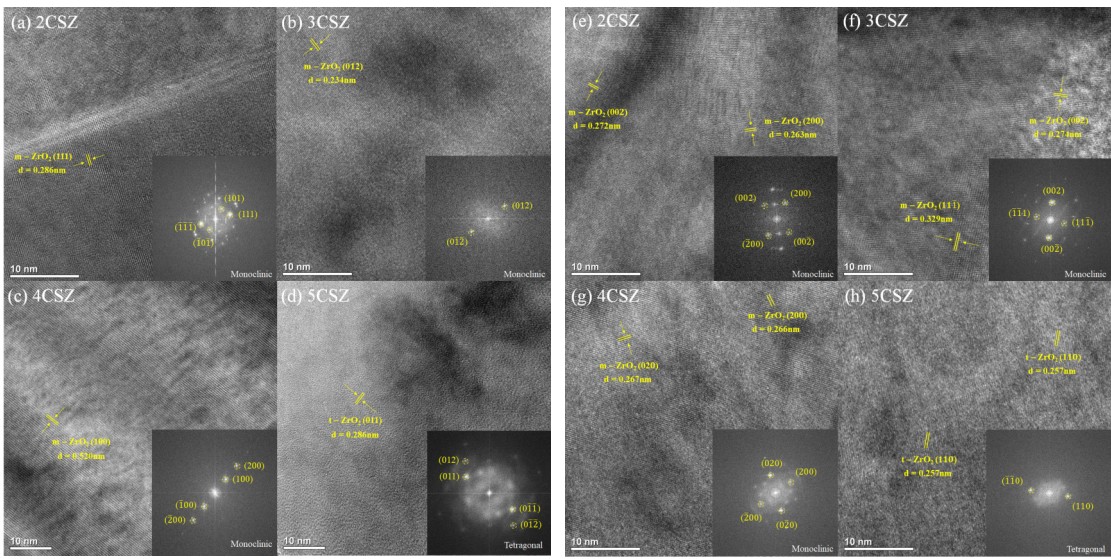

**Figure 2.** TEM images before (**a**–**d**) and after-heat treatment (**e**–**h**) of Ca-PSZ.

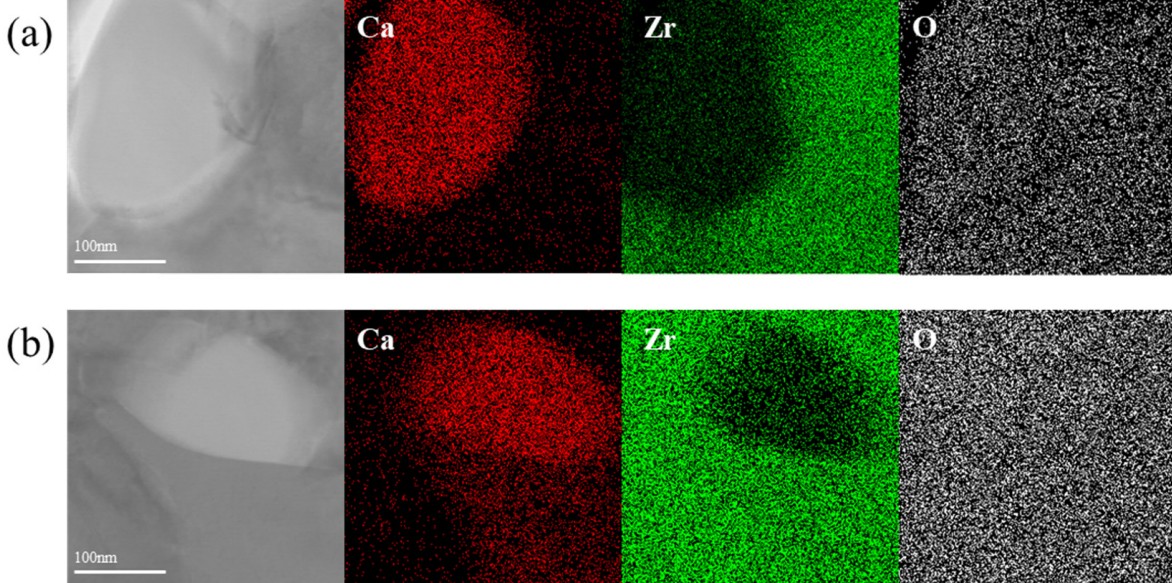

**Figure 3.** TEM-EDS images before (**a**) and after (**b**) heat treatment of 4CSZ.

Figure 4 shows the thermal expansion curve of the Ca-PSZ measured in the temperature range of 30–900 °C, and the coefficient of thermal expansion is listed in Table 3. As the CaO content increased, the CTE increased from $7.665 \times 10^{-6}$ K$^{-1}$ (2CSZ) to $8.811 \times 10^{-6}$ K$^{-1}$ (5CSZ). This is due to a decrease in the ratio of the monoclinic phase with a low CTE and an increase in the tetragonal phase with a high CTE [16]. Moreover, the increase in thermal expansion was interpreted as due to weakening binding energy in the crystal resulting from the increase of oxygen vacancies [17]. After the post-heat treatment, the CTE of Ca-PSZ became higher than that before the subsequent heat treatment. This indicates that the low-temperature monoclinic phase was stabilized into the high-temperature tetragonal phase as Ca was doped into the lattice owing to Ca diffusion [18].

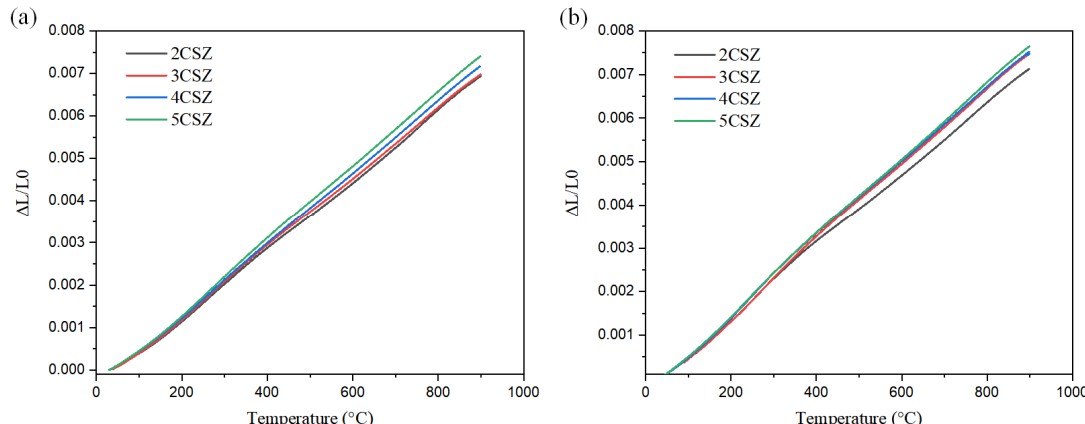

**Figure 4.** Thermal expansion curves before (**a**) and after-heat treatment (**b**) of Ca-PSZ.

**Table 3.** Thermal expansion coefficient of Ca-PSZ before-heat and after-heat treatment.

| Compound | Thermal Expansion Coefficient (1/K) | |
| :---: | :---: | :---: |
| | **Before-Heat Treatment** | **After-Heat Treatment** |
| 2CSZ | $7.665 \times 10^{-6}$ K$^{-1}$ | $8.049 \times 10^{-6}$ K$^{-1}$ |
| 3CSZ | $7.763 \times 10^{-6}$ K$^{-1}$ | $8.092 \times 10^{-6}$ K$^{-1}$ |
| 4CSZ | $8.583 \times 10^{-6}$ K$^{-1}$ | $8.605 \times 10^{-6}$ K$^{-1}$ |
| 5CSZ | $8.811 \times 10^{-6}$ K$^{-1}$ | $8.916 \times 10^{-6}$ K$^{-1}$ |

Figure 5 shows the Vickers hardness of Ca-PSZ. The hardness was improved in all the specimens as the amount of CaO increased, and in the case of 2CSZ, the hardness value was improved by approximately 35% after post-heat treatment. Moreover, in the case of 5CSZ, the highest hardness value was shown before and after post-heat treatment. This shows that the hardness was improved by transformation toughening in which the tetragonal phase transformed to the monoclinic phase when external energy was applied [16,19].

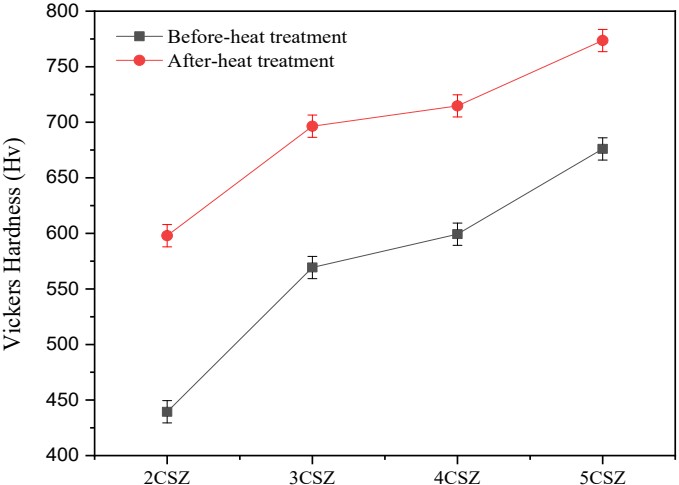

**Figure 5.** Vickers hardness before and after-heat treatment of Ca-PSZ.

## 4. Conclusions

We investigated phase formation and mechanical properties of Ca-PSZ by the post-heat treatment. The synthesized Ca-PSZ was partially stabilized by CaO as a mixed tetragonal and monoclinic phase as confirmed by the XRD pattern analysis. The intensity of the tetragonal ZrO$_2$ diffraction peak gradually increased and the intensity of the monoclinic ZrO$_2$ diffraction peak decreased after the post-heat treatment. In the TEM and the inverse

FFT images of Ca-PSZ, an increase in lattice parameter according to the CaO doping percentage was observed in the post heat-treated 2CSZ and 3CSZ $(002)_m$ plane d-spacing (0.272 nm and 0.274 nm). Additionally, the d-spacing of the 4CSZ $(200)_m$ plane increased from 0.260 nm to 0.266 nm after post-heat treatment, indicating the diffusion of $Ca^{2+}$ from rich regions to a lean region (monoclinic phase) after the post-heat treatment. The CTE increased from $7.665 \times 10^{-6}$ $K^{-1}$ (2CSZ) to $8.811 \times 10^{-6}$ $K^{-1}$ (5CSZ) as the CaO content increased. After the heat treatment, the CTE also increased from $8.049 \times 10^{-6}$ $K^{-1}$ (2CSZ) to $8.916 \times 10^{-6}$ $K^{-1}$ (5CSZ). The Vickers hardness of Ca-PSZ, and the hardness of all specimens improved as the amount of CaO increased. Furthermore, 5CSZ showed the largest hardness value after the post-heat treatment; specifically, the value increased by 14% after the treatment. This indicates that the low-temperature monoclinic phase was stabilized into the high-temperature tetragonal phase as Ca was dissolved in the lattice owing to Ca diffusion by the post-heat treatment.

**Author Contributions:** Conceptualization, H.L. (Heesoo Lee), H.Y., H.L. (Hwanseok Lee); investigation, H.Y., H.L. (Hwanseok Lee); methodology, analysis, H.L. (Heesoo Lee), H.Y., H.L. (Hwanseok Lee), K.J., J.K., I.J.; writing, H.L. (Heesoo Lee), H.Y., H.L. (Hwanseok Lee). All authors have read and agreed to the published version of the manuscript.

**Funding:** This work was supported by Korea Institute for Advancement of Technology (KIAT) grant funded by the Korea Government (MOTIE) (P0008335, The Competency Development Program for Industry specialist).

**Institutional Review Board Statement:** Not applicable.

**Informed Consent Statement:** Not applicable.

**Data Availability Statement:** Not applicable.

**Conflicts of Interest:** The authors declare no conflict of interest.

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
