# Peer review of "Phase Formation and Stabilization Behavior of Ca-PSZ by Post-Heat Treatment"

_metals, doi:10.3390/met12091479_

Round 1

Reviewer 1 Report

The presented paper deals with ceramic materials based on ZrO2, contained CaO. The powders were obtained by the mechanical mixing of the pure ZrO2 and CaO powders, which did not allow to perform the Ca-stabilized tetragonal ZrO2 phase. Thus, materials were formed by the predominantly monoclinic ZrO2 phase. These materials characterized by the low mechanical properties and do not find the broad application. The introduction requirs the more data on the Ca-stabilized ZrO2 powders synthesis and obtaining route, as well as results of the mechanical, thermal and phase evolution investigations of the similar Ca-stabilized ZrO2 materials. 

Also, the materials and methods should be improved by the description of the powders mixing procedure, as well as results of the powders properties investigations. The ZrO2 and CaO particle size and chemical composition distribution plays the important role on the ceramic properties. The dilatometric studies should be performed to evaluate the thermal behavior and thermal expansion reason.

Thus, the presented study should be noticeably improved before publication.

Author Response

Thank you for compliments and comments to the manuscript. Your sincere reviews have been very helpful to improve the quality of this paper.

Below is our reply to your comments.

[Reviewer’s comments]

Q 1) The presented paper deals with ceramic materials based on ZrO2, contained CaO. The powders were obtained by the mechanical mixing of the pure ZrO2 and CaO powders, which did not allow to perform the Ca-stabilized tetragonal ZrO2 phase. Thus, materials were formed by the predominantly monoclinic ZrO2 phase. These materials characterized by the low mechanical properties and do not find the broad application.

You will see that a number of general and specific points are mentioned which necessitate extensive rewriting of the paper. You will see that a number of general and specific points are mentioned which necessitate extensive rewriting of the paper.

Answer) Thank you for the comments. Various studies on yttria stabilized zirconia have been performed, but research on cost-effective partially stabilized zirconia for immersion nozzles such as Mg-PSZ and Ca-PSZ are lacking. We have conducted many studies on Mg-PSZ [Ref. 1-3], and in this study, we performed a study of Ca-PSZ materials for immersion nozzles. Ca-PSZ must have higher thermal stability and corrosion resistance to meet the more severe application conditions and higher requirements of continuous casting time. Doping other stabilizing elements to improve corrosion resistance have been studied [Ref. 4-5]. However, increasing the doping level is disadvantageous to the mechanical properties, so it is necessary to control the CaO content. In this study, when doped with CaO of lower mol% compared to the existing Ca-PSZ, the change of mechanical properties according to the post heat-treatment was studied. Corrosion resistance and mechanical properties when doped other stabilizing elements will be carried out in further study.

[1] Kim, B., & Lee, H. (2018). Valence state and ionic conduction in Mn‐doped MgO partially stabilized zirconia. Journal of the American Ceramic Society, 101(4), 1790-1795.

[2] Kim, B., Jeon, S., Erauw, J. P., & Lee, H. (2016). Scavenging effect on ionic conduction behaviour at the grain boundary of MgO partially stabilised zirconia. Advances in Applied Ceramics, 115(4), 200-203.

[3] Kim, B., Ryu, J., Jeon, S., & Lee, H. (2017). Oxygen vacancy generation and structural stability of MgO partially-stabilized zirconia upon MnO2 addition. Materials Research Express, 4(12), 126309.

[4] Volceanov, E., Abagiu, A., Becherescu, M., Volceanov, A., Niţă, P., Truşcă, R., & Mihalache, F. (2004). Development of zirconia composite ceramics and study on their corrosion resistance up to 1600 C. In Key Engineering Materials (Vol. 264, pp. 1739-1742). Trans Tech Publications Ltd.

[5] Jin, E., Yuan, L., Yu, J., Ding, D., & Xiao, G. (2022). Enhancement of thermal shock and slag corrosion resistance of MgO–ZrO2 ceramics by doping CeO2. Ceramics International, 48(10), 13987-13995.

Q 2) The introduction requires the more data on the Ca-stabilized ZrO2 powders synthesis and obtaining route, as well as results of the mechanical, thermal and phase evolution investigations of the similar Ca-stabilized ZrO2 materials. Materials and methods should be improved by the description of the powders mixing procedure, as well as results of the powders properties investigations. The ZrO2 and CaO particle size and chemical composition distribution plays the important role on the ceramic properties.

Answer) Thank you for the comments. Ball milling is known as a general method for synthesizing partially stabilized zirconia for structural materials. Since this study is not a thesis on the synthesis method, the contents of ball milling are described in detail in the experimental.

--------------------------------------------------------------------------------

Q 3) The dilatometric studies should be performed to evaluate the thermal behavior and thermal expansion reason.

Answer) Thank you for the comments. Spalling occurs due to thermal shock by molten steel in the case of immersion nozzle. Since coefficient of thermal expansion is one of the important factors for spalling, the dilatometric studies were performed.

Reviewer 2 Report

The manuscript can be accepted for publication in Metals.

A few minor issues  should be addressed:

1. In Table 2, first column 1, the order is wrong.

2. In Conclusion, line 133, "by the CaO  doping" should be deleted.

Author Response

Thank you for compliments and comments to the manuscript. Your sincere reviews have been very helpful to improve the quality of this paper.

Below is our reply to your comments.

[Reviewer’s comments]

Q 1) In Table 2, first column 1, the order is wrong.

You will see that a number of general and specific points are mentioned which necessitate extensive rewriting of the paper. You will see that a number of general and specific points are mentioned which necessitate extensive rewriting of the paper.

Answer) Thank you for the comment. We fixed the wrong order.

----------------------------------------------------------------------------------------------------------------

Q 2) In Conclusion, line 133, "by the CaO doping" should be deleted.

Answer) As the reviewer mentioned, we have revised the conclusion according to the reviewer’ comments.

Reviewer 3 Report

Title: appropriate and informative

Introduction – is necessary to expand the literature review for the Calcia partially stabilized zirconia previous work, even if it no so spread. However, by a quick search, I’ve found over 1000 papers in Science direct. I recommend discussing previously gained data by the scientific community and demonstrating the novelty of your work.

Experimental.

Regarding ball milling, it is an important step in powder preparation. Please include more details such as dry/wet milling was performed, balls and jar material (potentially it could contaminate the product), and milling time.

Table 1 why 2CSZ is bold & please remove the horizontal line under 2CSZ (the same for Table 3). By my opinion is useless to have a third column with Zr2/CaO (or change it to ZrO2 mol % and provide below the numbers), because it is clear by itself.

Please indicate how many parallel specimens and Vickers hardness tests for each were done.

Formula  (1) Please describe all variables

Results and discussion – in the common good, however, I would like to suggest comparing your results with others. Hope it will be possible. This is increased your paper's scientific value. 

Throughout the whole work please check spaces, somewhere it missing:

Table 3 "coefficient(1/K)" - should be space beforr "("

also please use the proper  symbol for temperature  - ° but not superscripted letter "o" 

Author Response

Thank you for compliments and comments to the manuscript. Your sincere reviews have been very helpful to improve the quality of this paper.

Below is our reply to your comments.

[Reviewer’s comments]

Q 1) Introduction is necessary to expand the literature review for the Calcia partially stabilized zirconia previous work, even if it no so spread. However, by a quick search, I’ve found over 1000 papers in Science direct. I recommend discussing previously gained data by the scientific community and demonstrating the novelty of your work.

You will see that a number of general and specific points are mentioned which necessitate extensive rewriting of the paper. You will see that a number of general and specific points are mentioned which necessitate extensive rewriting of the paper.

Answer) Thank you for the comment. We have faithfully revised the introduction to emphasize the novelty according to the reviewer’s comments.

Q 2) Regarding ball milling, it is an important step in powder preparation. Please include more details such as dry/wet milling was performed, balls and jar material (potentially it could contaminate the product), and milling time.

You will see that a number of general and specific points are mentioned which necessitate extensive rewriting of the paper. You will see that a number of general and specific points are mentioned which necessitate extensive rewriting of the paper.

Answer) Thank you for the comment. We included more details of ball milling experimental.

- Before

“The ZrO2 powders and the CaO powders were prepared as shown in Table 1. The powders were ball-milled and calcinated at 1000 °C for 2h.”

- After

“The ZrO2 powders and the CaO powders were prepared as shown in Table 1. The powders were ball-milled for 24 h in ethyl alcohol using partially stabilized zirconia balls and grinding pots. After drying, powders were calcinated at 1000 °C for 2 h.”

--------------------------------------------------------------------------------

Q 3) Table 1 why 2CSZ is bold & please remove the horizontal line under 2CSZ (the same for Table 3). By my opinion is useless to have a third column with ZrO2/CaO (or change it to ZrO2 mol % and provide below the numbers), because it is clear by itself.

You will see that a number of general and specific points are mentioned which necessitate extensive rewriting of the paper. You will see that a number of general and specific points are mentioned which necessitate extensive rewriting of the paper.

Answer) Thank you for the comment. We have revised the Table 1 according to the reviewer’s comments.

--------------------------------------------------------------------------------

Q 4) Indicate how many parallel specimens and Vickers hardness tests for each were done.

Answer) As the reviewer mentioned, we added how many parallel specimens and Vickers hardness tests for each were done

--------------------------------------------------------------------------------

Q 5) Formula (1) Please describe all variables

You will see that a number of general and specific points are mentioned which necessitate extensive rewriting of the paper. You will see that a number of general and specific points are mentioned which necessitate extensive rewriting of the paper.

Answer) Thank you for the comment. We have described all variables in the formula (1) according to the reviewer’s comments.

--------------------------------------------------------------------------------

Q 6) In the common good, however, I would like to suggest comparing your results with others. Hope it will be possible. This is increased your paper's scientific value.

Answer) As the reviewer mentioned, we modified results and discussion section.

--------------------------------------------------------------------------------

Q 7) Table 3 "coefficient(1/K)" - should be space beforr "(" also please use the proper symbol for temperature - ° but not superscripted letter "o"

Answer) As the reviewer mentioned, we fixed table 3 and used the proper symbol for temperature.

Reviewer 4 Report

P5 The lines in the Fig 4 is not clear, please change it.

Author Response

Thank you for compliments and comments to the manuscript. Your sincere reviews have been very helpful to improve the quality of this paper.

Below is our reply to your comments.

[Reviewer’s comments]

Q 1) P5 The lines in the Fig 4 is not clear, please change it.

You will see that a number of general and specific points are mentioned which necessitate extensive rewriting of the paper. You will see that a number of general and specific points are mentioned which necessitate extensive rewriting of the paper.

Answer) Thank you for the comments. To make clear the lines in Figure, we replaced Figure 4.

Round 2

Reviewer 3 Report

Thank you for the improvements.